# Qualitative and Quantitative Quality Assessment of Low-Light Enhanced Images: A Dataset and Benchmark Metric

## Abstract

Low-light image enhancement (LLIE) improves visibility and restores details in challenging lighting conditions. It is crucial to fairly evaluate LLIE methods to foster the development of more effective models. However, quality assessment of low-light enhanced images proves to be as challenging as the enhancement itself. From a quantitative perspective, full-reference image quality assessment (FR-IQA) metrics (e.g., PSNR and SSIM) are commonly employed to assess the perceptual quality of enhanced images. However, they are not suitable when a pristine reference image is unavailable, which is often the case in real-world applications. From a qualitative perspective, the absence of a standardized and reproducible evaluation pipeline makes it extremely difficult to ensure fair comparisons across different studies. To confront these challenges, we present the Low-light Image Distortions and Quality (LIDQ) dataset, featuring both overall quality scores and distortion distribution annotations collected through formal subjective testing. Leveraging LIDQ, we propose a no-reference Low-light Enhanced Image Quantitative and Qualitative Quality Assessment (LIQ$^3$A) method that not only estimates perceptual quality without requiring a reference, but also provides qualitative assessments of enhancement-induced distortions. Experiments show that LIQ$^3$A aligns closely with human perception while accurately identifying distortion patterns. We anticipate that the proposed dataset and metric will facilitate future advances in low-light image enhancement by providing reliable evaluation feedback.

## 1 Introduction

Images captured in low-light environments frequently suffer from visual degradations, e.g., poor visibility, low contrast, and severe noise, which can significantly compromise visual perception and hinder the performance of computer vision tasks (Yang et al., 2020). Although advancements in imaging hardware and specialized photographic techniques can partially alleviate these issues, they often fail to completely eliminate noise due to the limited light available to camera sensors. Increasing exposure time may reduce noise but frequently introduces motion blur, further deteriorating image quality (Wang et al., 2023c). As a cost-effective alternative, computational low-light image enhancement (LLIE) methods have gained considerable attention (Guo et al., 2016). These LLIE methods focus on improving visibility, enhancing contrast, and suppressing noise, rendering images with higher perceptual quality, and boosting downstream applications' performance (Guo et al., 2020).

Despite recent advances, LLIE methods still produce artifacts such as amplified noise, color distortions, and over-smoothing that compromise image quality (Wang et al., 2024a). Assessing the perceptual quality of enhanced images is thus critical for evaluating LLIE performance and guiding refinement, typically through quantitative and qualitative evaluations (Chen et al., 2023; Zhai et al., 2021). Quantitative assessment employs image quality assessment (IQA) methods, which assign scalar values to enhancement performance. Depending on reference availability, IQA methods are categorized as full-reference, reduced-reference, or no-reference (blind) (Zhang et al., 2023b). Blind IQA (BIQA) is particularly practical since it does not require pristine references, which are often unavailable in real-world scenarios (Mittal et al., 2012b). Qualitative assessments, by contrast, rely on visual inspection to reveal strengths and weaknesses more intuitively than a single score, but they

Table 1: Summary of the previous IQA datasets for low-light (enhanced) images. 2AFC: Two-alternative forced choice. SS: Single stimulus. DS: Double stimulus. Con.: Contrast. Alg.: Algorithm. ACJ: Adjectival categorical judgement. CQR: Continuous quality rating. QSD: Quality semantic description. DTSR: Distortion types and severity ratings.

| Dataset | # Reference images | Enhancement types | # Enhancement methods | # Image | # Subjects | Judgment type |
|---|---|---|---|---|---|---|
| Chen14 (Chen et al., 2014) | 100 | Alg. outputs | 5 | 500 | - | 2AFC |
| CCID2014 (Gu et al., 2015) | 15 | Con.-enhanced | 5 | 655 | 22 | SS-CQR |
| NNID (Xiang et al., 2019) | 448 | Real-captured | - | 2240 | 74 | SS-ACJ |
| LIEQ (Zhai et al., 2021) | 100 | Alg. outputs | 10 | 1,000 | 21 | SS-CQR |
| LEISD (Lin et al., 2023) | 255 | Alg. outputs | 8 | 2,040 | 20 | SS-CQR |
| EHNQ (Yang et al., 2023b) | 100 | Alg. outputs | 15 | 1,500 | 50 | DS-ACJ |
| SQUARE-LOL (Chen et al., 2023) | 290 | Alg. outputs | 10 | 2,900 | 30 | 2AFC |
| RNTIEQA (Wang et al., 2024b) | 200 | Alg. outputs | 10 | 2,000 | 15 | 2AFC |
| MLIQ (Wang et al., 2024a) | 1,360 | Real-captured | - | 1,360 | 26 | SS-CQR & QSD |
| LIDQ (Ours) | 253 | Alg. outputs | 22 | 5,566 | 34 | SS-ACJ & DTSR |

are usually limited to a small sample set, leading to sample bias or the so-called *cherry-picking* issue (Cao et al., 2021). Existing BIQA metrics further lack support for such qualitative comparisons, hindering large-scale dataset-level benchmarking.

In this work, we reformulate qualitative assessment as the estimation of distortion distributions in enhanced images. This approach makes the qualitative assessment process quantifiable, enabling evaluations on full datasets and ensuring the reproducibility of enhanced images across different studies. To this end, we introduce the Low-light Image Distortions and Quality (LIDQ) dataset, comprising $(21+1) \times 253 = 5,566$ images with the most comprehensive quantitative and qualitative annotations to date. Specifically, we assemble a total of 253 distinct low-light images from existing paired LLIE datasets to serve as reference inputs for enhancement, each accompanied by 1 normal-light ground truth. We then enhance these reference images utilizing over 21 state-of-the-art LLIE methods, resulting in the collection of 5,566 images. A comprehensive subjective quality assessment is conducted to gather human quantitative mean opinion scores (MOSs) of image quality, alongside qualitative annotations of enhancement-induced distortions.

We also leverage the LIDQ dataset to develop the blind Low-light Enhanced Image Quantitative and Qualitative Quality Assessment (LIQ³A) model, a strong baseline that evaluates the quality of low-light enhanced images from both quantitative and qualitative perspectives, without relying on pristine ground-truths. Adopting a multitask learning framework, LIQ³A seamlessly integrates qualitative insights into BIQA learning. The model is trained to simultaneously predict quantitative quality scores and estimate qualitative distortion patterns. Built on a pretrained vision-language model, LIQ³A bridges the two tasks using textual templates. By computing a joint probability based on the cosine similarities between visual and textual embeddings, the model makes predictions for both tasks and optimizes them through carefully designed loss functions.

Overall, our contributions are threefold:

- We establish LIDQ, a comprehensive quality assessment dataset consisting of 5,566 annotated images. Both quantitative quality ratings and qualitative annotations are collected for each image through formal subjective testing.
- Based on LIDQ, we propose LIQ³A, a computational quality metric that assesses low-light enhanced images from both quantitative and qualitative perspectives.
- Extensive experiments on multiple datasets show that LIQ³A achieves closer alignment with human quantitative annotations than other BIQA methods and effectively identifies distortion patterns in enhanced images.

## 2 RELATED WORKS

**Low-light Image Enhancement**. Classical LLIE methods, such as histogram equalization (Pizer et al., 1987) and Retinex-based techniques (Wei et al., 2018), rely on handcrafted priors and complex optimization, often resulting in limited performance or high computational cost (Li et al., 2015; Ying et al., 2017; Zheng et al., 2022). In contrast, deep neural network (DNN)-based LLIE approaches automatically learn features from data and enhance brightness, contrast, and detail via end-to-end training (Lore et al., 2017). These models are typically trained on paired datasets—either synthetic or

real—though data collection is often costly (Zheng et al., 2022; Zhang et al., 2019; 2021b). To address this, alternative learning strategies such as unsupervised (Jiang et al., 2021), semi-supervised (Yang et al., 2021a), and zero-shot learning (Zhang et al., 2024) have emerged. Despite their effectiveness, LLIE methods may still introduce distortions like structural artifacts, color shifts, or noise (Zhai et al., 2021), underscoring the need for reliable quality assessment.

**Blind Image Quality Assessment**. BIQA estimates perceptual image quality without reference images, providing an efficient alternative to subjective testing (Mittal et al., 2012b). Early methods relied on handcrafted features (Mittal et al., 2012a) but lacked robustness across diverse content and distortions. Deep learning greatly improved BIQA by modeling complex content–distortion interactions (Zhang et al., 2021a; Ke et al., 2021), and recent multimodal vision–language approaches further enhance performance (Zhang et al., 2023b; Wu et al., 2024b). However, most BIQA methods are designed for synthetic or real-captured distortions, which differ from those introduced by low-light enhancement. LLIE often produces seemingly realistic but fabricated textures (Gu et al., 2020), leading existing BIQA models to generalize poorly and suffer notable performance drops (Wang et al., 2024a). Therefore, developing BIQA methods tailored specifically for LLIE is essential.

**Low-light Enhanced Image Quality Assessment**. Traditional LLIE methods primarily focus on brightness adjustment, and accordingly, assessment metrics often rely on handcrafted features that capture brightness or color-related characteristics Lin et al. (2023). With the rise of DNNs, learning-based BIQA methods for LLIE have emerged, enabling models to capture not only luminance-related cues but also complex, hard-to-model artifact patterns Chen et al. (2014); Gu et al. (2017). More recently, multimodal approaches have expanded the form of annotations used for quality assessment, for example by leveraging image–text information to improve perceptual quality prediction. However, these BIQA methods still provide only scalar quantitative scores and lack quantifiable distortion analysis Wu et al. (2024a). Incorporating qualitative insights is crucial for advancing LLIE assessment and guiding the development of more robust enhancement models.

**Low-light IQA Datasets**. Recent years have seen growing efforts to develop datasets tailored for LLIE evaluation, spanning perceptual-quality benchmarks (see Table 1) and high-level task datasets such as ExDark (Loh & Chan, 2019). In this work, we focus specifically on perceptual assessment. Most IQA datasets focus on algorithm-enhanced images, whereas NNID (Xiang et al., 2019) and MLIQ (Wang et al., 2024a) evaluate real-captured low-light images. Most datasets provide only quantitative scores such as MOS, which merge multiple distortions into a single value and lack explicit qualitative descriptions of LLIE-induced artifacts (e.g., amplified noise, color shifts, blur). As a result, evaluation largely remains quantitative, while qualitative analysis is restricted to visually checking a few samples—leading to the well-known *cherry-picking* issue. LIDQ addresses this gap by offering paired quantitative and qualitative labels for every enhanced image. Although MLIQ includes semantic descriptions, it focuses on device-induced degradations and uses free-form text not aligned with controlled distortion categories, limiting fine-grained analysis of LLIE-specific artifacts.

## 3 PROPOSED DATASET: LIDQ

**Reference Image Collection**. We compile 253 low-light images for enhancement, including 15 from the LOL-v1 test set (Wei et al., 2018), 100 from the LOL-v2 test set (Yang et al., 2021b), and 138 frames from the SDSD dataset (Wang et al., 2021), where one representative frame is extracted from each video. These datasets are selected because they cover diverse indoor and outdoor scenes and are widely recognized benchmarks in the LLIE literature (Yan et al., 2025), frequently adopted to enable fair and consistent comparisons. Regarding the overlapping in the LOL datasets, we leverage the shared samples between LOL-v1 and LOL-v2 to verify the consistency of subjective ratings, ensuring that no overlapping samples appear simultaneously across the training, validation, or test splits.

**Low-Light Enhancement Methods**. We employ 21 recent LLIE methods[1] (see Figure 1 (a)) to generate enhanced images. These methods span Retinex-based, flow-based, Transformer-based,

---

[1]These methods include Zero-DCE (Guo et al., 2020), EnlightenGAN (Jiang et al., 2021), RUAS (Risheng et al., 2021), SCI (Ma et al., 2022), GSAD (Hou et al., 2023), PairLIE (Fu et al., 2023), NeRCo (Yang et al., 2023a), CLIP-LIT (Liang et al., 2023), AGLLNet (Lv et al., 2021), RQ-LLIE (Liu et al., 2023), MBLLEN (Lv et al., 2018), RetinexNet (Wei et al., 2018), KinD++ (Zhang et al., 2019), MIRNet (Zamir et al., 2020), SNR-Net (Xu et al., 2022), IAT (Cui et al., 2022), LLFlow (Wang et al., 2022), Retinexformer (Cai et al., 2023), RetinexMamba (Bai et al., 2024), LLFormer (Wang et al., 2023b), SCUNet (Zhang et al., 2023a).

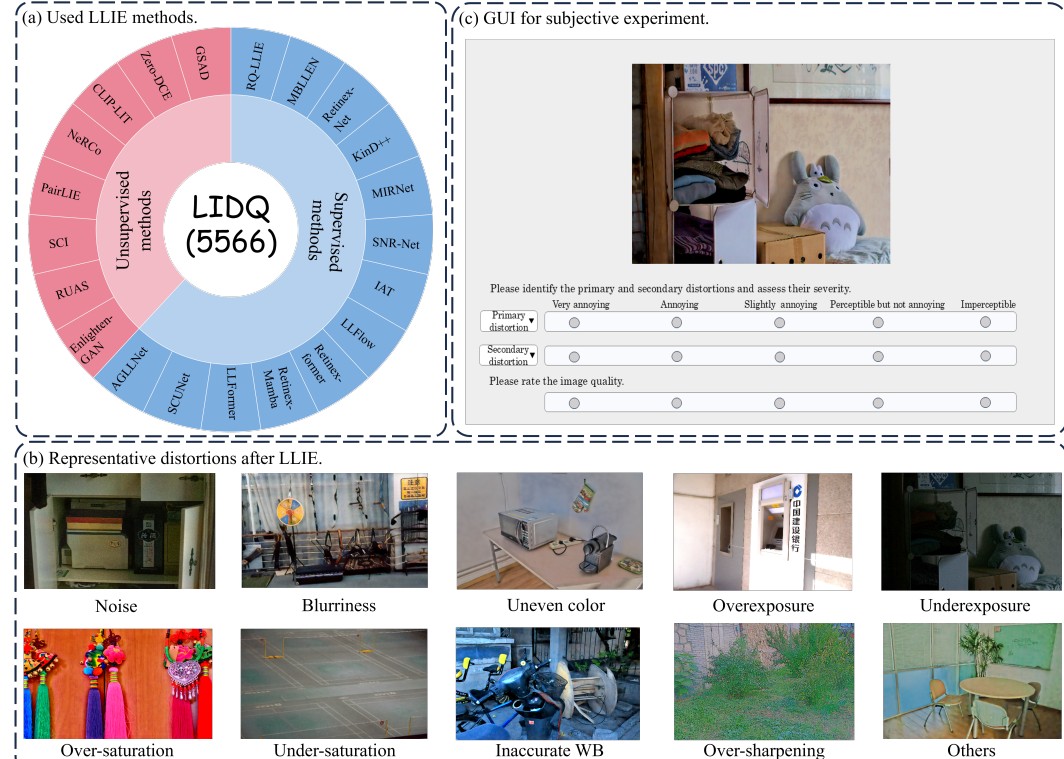

Figure 1: LIDQ comprises enhanced images generated by various LLIE algorithms (see (a)), featuring diverse algorithm-dependent artifacts (see (b)). Both quality and distortion annotations are obtained through formal subjective testing using the graphical interface shown in (c).

GAN-based, and zero-reference approaches, providing a broad spectrum of distortion behaviors. For methods such as Zero-DCE, EnlightenGAN, RUAS, SCI, GDP, PairLIE, NeRCo, and CLIP-LIT, we adopt the official models released by the authors. For the remaining methods, we use publicly available models trained on LOL-v2 or retrain them following the authors' default configurations. This process produces 5,313 enhanced images, and with the corresponding ground truth from the source datasets, yields a total of 5,566 annotated images.

**Subjective Testing**. We design a graphical user interface (GUI) (see Figure 1(b)) to collect both quantitative and qualitative annotations. Prior to the experiment, subjects are instructed to perform the evaluation on high-resolution monitors and are provided with detailed guidelines, including the definition of "technical image quality," examples of common distortion types. For quantitative assessment, we adopt the standard 5-point ACJ scale (Hosu et al., 2020). For qualitative evaluation, subjects identify the primary and secondary salient distortions from nine categories[2], and then rate the severity of each selected distortion using a 5-point ACJ scale. Compared with LOL-v1/v2, SDSD images exhibit stronger low-light noise and blur. To account for this, we perform separate subjective evaluations for enhanced images from each dataset, producing two distinct subsets.

Table 2: Min, max, median and mean SRCC, and PLCC between two randomized subgroups with equal size across 100 splits.

| Criterion | Min | Max | Median | Mean |
|---|---|---|---|---|
| SRCC ↑ | 0.792 | 0.901 | 0.870 | 0.868 |
| PLCC ↑ | 0.794 | 0.898 | 0.870 | 0.869 |

We recruited 34 subjects with normal or corrected-to-normal vision and verified color perception using the Ishihara test (Wang et al., 2023d). Each subject evaluated a subset of images, with every image annotated by at least 15 subjects (Wang et al., 2024a; Hosu et al., 2020). The assessment started with quantifiable distortion severity annotations, followed by overall quality ratings (Fang et al., 2020). In total, we collected 89,454 annotations across 5,566 images.

---

[2]These distortions include noise, blurriness, uneven color, overexposure, underexposure, over-saturation, under-saturation, inaccurate white balance, over-sharpening, and others (Wang et al., 2024a; Lin et al., 2023).

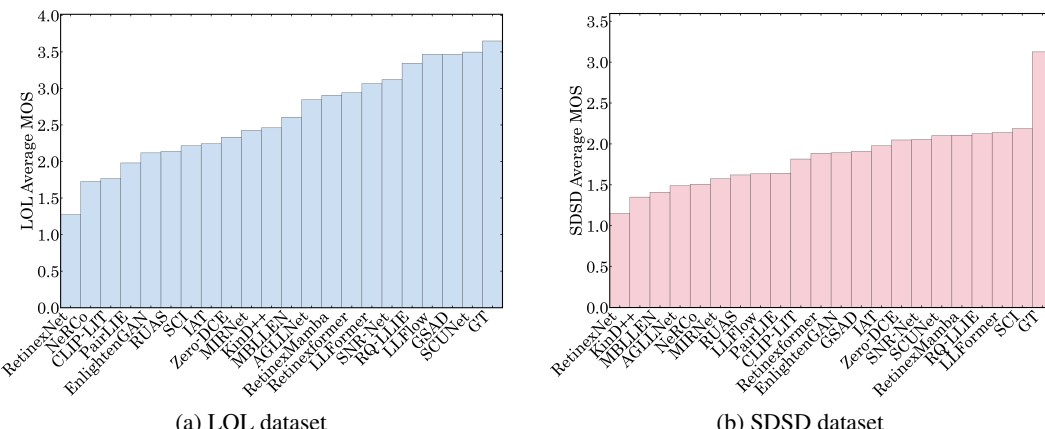

(a) LOL dataset          (b) SDSD dataset

Figure 2: The average MOS scores of enhanced images produced by different LLIE methods on the LOL and SDSD subsets, sorted in ascending order.

**Subjective Data Processing**. We apply rigorous outlier and subject filtering based on the ITU-R BT.500-13 methodology (BT.500 ITU-R, 2002) to ensure annotation reliability. Annotations deviating by more than three standard deviations from the mean are marked as outliers, and subjects with outlier rates above 5% are excluded. After filtering, all subjects remain valid, and 2.61% of ratings are discarded as outliers. The mean of the remaining valid ratings is used as the ground-truth MOS. For qualitative annotations, all subjects' primary and secondary selections are pooled to form a probabilistic distortion distribution. Because subjects may focus on different artifacts, the aggregated distribution captures both dominant distortions (with higher probability) and less salient or mixed distortions (with lower probability). We then convert the selected distortion types and their severity scores into a continuous probability vector over all categories.

**Subjective Results and Analysis**. To assess annotation reliability, we repeatedly split subjects into two subgroups and computed the Spearman rank correlation coefficient (SRCC) and Pearson linear correlation coefficient (PLCC) between their mean MOS ratings. Averaged over 100 trials, both metrics exceeded 0.85 (see Table 2), indicating strong rating consistency. Fig. 2 shows the average MOS for each LLIE method, leading to several key observations. First, GT images consistently exhibit higher perceptual quality than all enhanced versions, indicating significant room for improvement. Second, the quality gap between GT and enhanced images is notably larger in the SDSD subset than in LOL, highlighting the greater challenges of enhancing low-light images in the wild. Third, the relative rankings of different LLIE algorithms vary across the two subsets. For example, SCI (Ma et al., 2022) ranks highest among all LLIE algorithms on the SDSD subset, but its ranking significantly drops on the LOL subset. This highlights the limited generalizability of LLIE models and the need for diverse evaluation to ensure reliable performance across varied scenarios. We show the distortion distributions of enhanced images corresponding to all LLIE methods in Fig. 3. It is evident that noise and blurriness are the most common artifacts in enhanced low-light images, followed by underexposure and white balance (WB) issues, highlighting key challenges faced by current LLIE methods. Fig. 3 also provides an intuitive overview of the relative strengths and weaknesses of various LLIE methods, e.g., NeRCo and SNR-Net show stronger resistance to noise artifacts but are more susceptible to blurriness.

## 4   PROPOSED METRIC: LIQ³A

**Preliminaries**. Given an image $\mathbf{x} \in \mathbb{R}^N$ produced by a LLIE method, LIQ³A is designed to estimate the perceptual quality of $\mathbf{x}$ through a mapping $\hat{q} : \mathbb{R}^N \to \mathbb{R}$, and to characterize the distortion profile of $\mathbf{x}$ by converting the qualitative analysis into a distribution over $M$ candidate distortion types via a mapping $\hat{d} : \mathbb{R}^N \to \mathbb{R}^M$. For quality prediction, we adopt a five-level Likert scale $\mathcal{C} = \{1, 2, 3, 4, 5\}$ ({"bad", "poor", "fair", "good", "perfect"}) and define the predicted quality score $\hat{q}$ as

$$\hat{q}(\mathbf{x}) = \sum_{c=1}^{C} \hat{p}(c \mid \mathbf{x}) \times c, \tag{1}$$

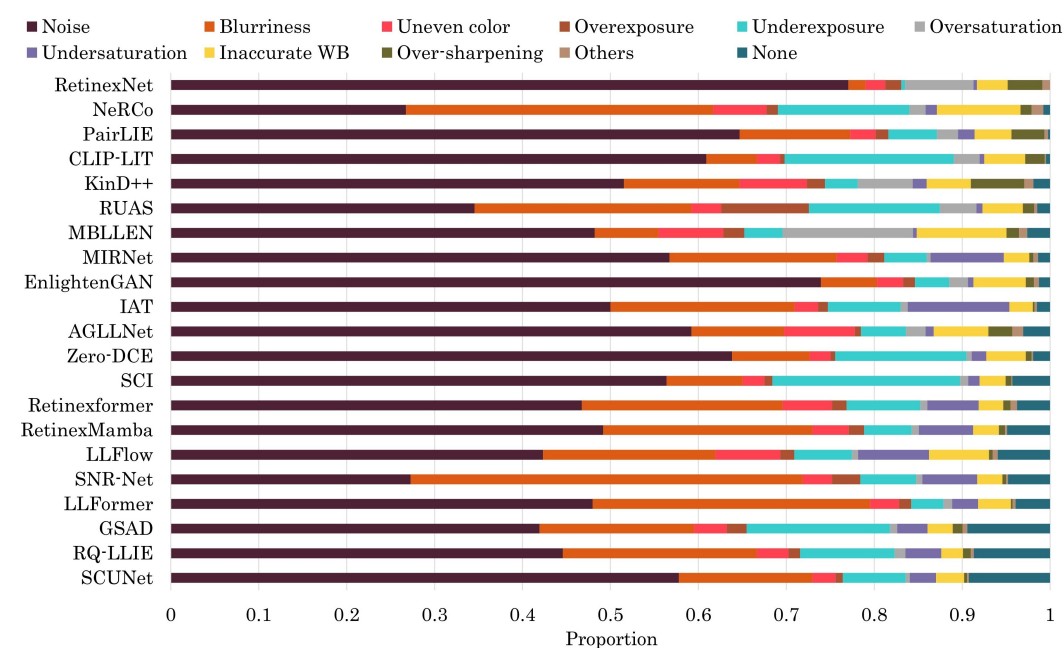

Figure 3: The distortion distributions of all LLIE methods on the LIDQ dataset.

where $C = 5$ and $\hat{p}(c \mid \mathbf{x})$ denote the estimated marginal probability of level $c$. As for distortion analysis, we consider $M = 11$ distortion types as specified in Sec. 3. To build LIQ³A, we leverage the strong representational power of a CLIP-style (Radford et al., 2021) vision-language model, SigLIP-2 (Tschannen et al., 2025), pre-trained on 12 billion image-text pairs. We utilize the built-in language module of SigLIP-2 to bridge the two tasks by constructing a textual template that combines labels from both: "*a photo with {d} artifacts, which is of {c} quality*", yielding $5 \times 11 = 55$ textual descriptions.

**Model Specification**. We use the NaFlex variant of SigLIP-2, which inherently supports multi-resolution inputs while preserving aspect ratios. Given a patch size and a target sequence length, NaFlex resizes input images to dimensions that are multiples of the patch size, minimizing aspect ratio distortion while ensuring the sequence length remains within bounds. The model comprises a visual encoder $\boldsymbol{f_\phi} : \mathbb{R}^N \to \mathbb{R}^K$ and a language encoder $\boldsymbol{g_\varphi} : \mathcal{T} \to \mathbb{R}^K$, parameterized by $\phi$ and $\varphi$, respectively, where $\mathcal{T}$ denotes the text prompt set. Thanks to the NaFlex mechanism, we can efficiently extract multi-scale visual representations. Each input image $\boldsymbol{x}$ is resized to $U$ different resolutions, from which we derive the visual embedding matrix $\boldsymbol{F}(\boldsymbol{x}) \in \mathbb{R}^{U \times K}$. In parallel, we encode $V = 55$ candidate text prompts to obtain the textual embedding matrix $\boldsymbol{G}(\boldsymbol{x}) \in \mathbb{R}^{V \times K}$.

We then compute the cosine similarity between the visual embedding of the $u$-th image $\boldsymbol{F}_{u\bullet}$ (as a row vector and for $1 \leq u \leq U$) and $v$-th candidate textual embedding $\boldsymbol{G}_{v\bullet}$ (corresponding to a particular set of $\{c, d\}$), averaging across $U$ sub-images to obtain the image-level correspondence score:

$$\text{logit}(c, d|\boldsymbol{x}) = \frac{1}{U} \sum_{u=1}^{U} \frac{\boldsymbol{F}_{u\bullet}(\boldsymbol{x})\boldsymbol{G}_{v\bullet}^{\mathsf{T}}(\boldsymbol{x})}{\|\boldsymbol{F}_{u\bullet}(\boldsymbol{x})\|_2 \|\boldsymbol{G}_{v\bullet}(\boldsymbol{x})\|_2}. \tag{2}$$

After matching the image with all candidate descriptions, we apply a softmax with learnable temperature $\tau$ and bias $\beta$ to compute the joint probability:

$$\hat{p}(c, d \mid \boldsymbol{x}) = \frac{\exp\left(\text{logit}(c, d \mid \boldsymbol{x})/\tau + \beta\right)}{\sum_{c,d} \exp\left(\text{logit}(c, d \mid \boldsymbol{x})/\tau + \beta\right)}. \tag{3}$$

**Loss for Quantitative Assessment**. We marginalize $\hat{p}(c, d|\boldsymbol{x})$ to compute $\hat{p}(c|\boldsymbol{x})$, from which we obtain the quality estimate $\hat{q}(\boldsymbol{x}) \in \mathbb{R}$ by Eq. equation 1. During training, we sample a mini-batch $\mathcal{B} = \{\boldsymbol{x}_i, q(\boldsymbol{x_i})\}_{i=1}^{|\mathcal{B}|}$ at each iteration, where $q(\boldsymbol{x_i})$ is the MOS of $\boldsymbol{x}_i$. We compute a binary label

indicating the relative quality ranking of two images within $\mathcal{B}$:

$$p(\boldsymbol{x}_i, \boldsymbol{x}_j) = \begin{cases} 1 & \text{if } q(\boldsymbol{x}_i) \geq q(\boldsymbol{x}_j) \\ 0 & \text{otherwise} \end{cases}, \tag{4}$$

Following Thurstone's model (Thurstone, 1927), we estimate the probability that $\boldsymbol{x}_i$ is perceived as better than $\boldsymbol{x}_j$ by:

$$\hat{p}(\boldsymbol{x}_i, \boldsymbol{x}_j) = \Phi\left(\frac{\hat{q}(\boldsymbol{x}_i) - \hat{q}(\boldsymbol{x}_j)}{\sqrt{2}}\right), \tag{5}$$

where $\Phi(\cdot)$ is the cumulative distribution function of a standard normal distribution. We adopt the fidelity loss (Tsai et al., 2007) as the statistical distance measure:

$$\ell_f(\mathcal{B}) = \frac{1}{|\mathcal{B}|} \sum_{\{(x_i, x_j), p\} \in \mathcal{B}} \left(1 - \sqrt{p(\boldsymbol{x}_i, \boldsymbol{x}_j)\hat{p}(\boldsymbol{x}_i, \boldsymbol{x}_j)} - \sqrt{(1 - p(\boldsymbol{x}_i, \boldsymbol{x}_j))(1 - \hat{p}(\boldsymbol{x}_i, \boldsymbol{x}_j))}\right). \tag{6}$$

To improve the precision of quantitative quality prediction, we incorporate an additional loss term based on the PLCC:

$$\ell_p = 1 - \frac{\sum_{i=1}^{|\mathcal{B}|} \left(\hat{q}(\boldsymbol{x}_i) - \overline{\hat{q}}\right)\left(q(\boldsymbol{x}_i) - \overline{q}\right)}{\sqrt{\sum_{i=1}^{|\mathcal{B}|}\left(\hat{q}(\boldsymbol{x}_i) - \overline{\hat{q}}\right)^2}\sqrt{\sum_{i=1}^{|\mathcal{B}|}\left(q(\boldsymbol{x}_i) - \overline{q}\right)^2}}. \tag{7}$$

where $\overline{q} = \frac{1}{|\mathcal{B}|}\sum_{i=1}^{|\mathcal{B}|} q(\boldsymbol{x}_i)$ and $\overline{\hat{q}} = \frac{1}{|\mathcal{B}|}\sum_{i=1}^{|\mathcal{B}|} \hat{q}(\boldsymbol{x}_i)$.

**Loss for Qualitative Assessment**. Given $\hat{p}(c, d|\boldsymbol{x})$, we marginalize it to obtain $\hat{p}(d|\boldsymbol{x})$. We again use the fidelity loss to measure the distance between the predicted and ground-truth distortion distributions $p(d|\boldsymbol{x}) \in \mathbb{R}^M$:

$$\ell_d(\boldsymbol{x}) = \frac{1}{|\mathcal{B}|} \sum_{\boldsymbol{x} \in \mathcal{B}} \left(1 - \sqrt{p(d|\boldsymbol{x})\hat{p}(d|\boldsymbol{x})} - \sqrt{(1 - p(d|\boldsymbol{x}))(1 - \hat{p}(d|\boldsymbol{x}))}\right). \tag{8}$$

Finally, we compute the overall loss as: $\ell = \lambda_f \ell_f + \lambda_p \ell_p + \lambda_d \ell_d$, where $\lambda_f$, $\lambda_p$, and $\lambda_d$ are weighting factors that balance the contribution of each loss term.

## 5 EXPERIMENTS AND RESULTS

**Experimental Setups**. We conduct experiments on both subsets of the LIDQ dataset. To ensure meaningful supervision, we perform joint training on both subsets using a pairwise learning-to-rank strategy (Zhang et al., 2021a) restricted to within-subset comparisons. Each of Subset 1 and Subset 2 is partitioned into training, validation, and testing splits in a 7:1:2 ratio, ensuring that visually similar content is confined to a single split to avoid content leakage. Our model is instantiated using the SigLIP-2-base-NaFlex variant (Tschannen et al., 2025), which employs a shared ViT-B/16 architecture (Dosovitskiy et al., 2021) for both visual and language encoders. The visual encoder leverages the NaFlex mechanism to preprocess inputs in an aspect-ratio-preserving manner, based on a preset maximum patch count. We adopt three such settings ($U = 3$): 196, 529, and 1024 patches. The language encoder processes tokenized text truncated to the first 64 tokens, using the Gemma tokenizer (Team et al., 2024). Training is conducted using the AdamW optimizer (Loshchilov & Hutter, 2019) with a weight decay of $10^{-3}$ and an initial learning rate of $5 \times 10^{-6}$, scheduled via cosine annealing (Loshchilov & Hutter, 2017). We train LIQ³A for 8 epochs with a mini-batch size of 16 for each subset. All loss weights, i.e., $\lambda_f$, $\lambda_p$, and $\lambda_d$, are set to 1. We use SRCC and PLCC as prediction monotonicity and precision measures, respectively. Additionally, the Earth Mover's Distance (EMD) (Levina & Bickel, 2001) is utilized to measure the closeness between the predicted and ground-truth distortion distributions.

**Comparison Methods**. We compare the performance of the proposed LIQ³A with eleven BIQA methods, including five pre-trained models—MUSIQ (Ke et al., 2021), CLIPIQA (Wang et al., 2023a), QualiCLIP+ (Agnolucci et al., 2024a), UNIQUE (Zhang et al., 2021a), and VisualQuality-R1 (Wu et al., 2025)—as well as seven models re-trained on LIDQ[3]: DBCNN (Zhang et al., 2018),

---
[3]These models are re-trained and evaluated using the same data splits as LIQ³A.

Table 3: Performance comparison of different IQA algorithms on Subset 1 and Subset 2 (↑ means higher is better, ↓ means lower is better).

| Method | Subset 1 (LOL-v1 & LOL-v2) | | | Subset 2 (SDSD) | | |
|---|---|---|---|---|---|---|
| | SRCC (↑) | PLCC (↑) | EMD (↓) | SRCC (↑) | PLCC (↑) | EMD (↓) |
| MUSIQ (Ke et al., 2021) | 0.6871 | 0.7017 | – | 0.4643 | 0.5517 | – |
| CLIPIQA (Wang et al., 2023a) | 0.1748 | 0.2190 | – | 0.3929 | 0.4124 | – |
| QualiCLIP+ (Agnolucci et al., 2024a) | 0.5788 | 0.6060 | – | 0.5952 | 0.7132 | – |
| UNIQUE (Zhang et al., 2021a) | 0.5881 | 0.6109 | – | 0.4004 | 0.5409 | – |
| VisualQuality-R1 (Wu et al., 2025) | 0.6897 | 0.7052 | – | 0.6346 | 0.6314 | – |
| DBCNN (Zhang et al., 2018) | 0.8072 | 0.8267 | – | 0.9284 | 0.9078 | – |
| HyperIQA (Su et al., 2020) | 0.7201 | 0.7385 | – | 0.9100 | 0.9231 | – |
| MANIQA (Yang et al., 2022) | 0.8047 | 0.8260 | – | 0.9213 | 0.9354 | – |
| ARNIQA (Agnolucci et al., 2024b) | 0.7335 | 0.7472 | – | 0.9159 | 0.9260 | – |
| TOPIQ (Chen et al., 2024) | 0.7563 | 0.7659 | – | 0.9210 | 0.9355 | – |
| Q-Align Wu et al. (2024c) | 0.8410 | 0.8512 | – | 0.9062 | 0.9047 | – |
| LIQE (Zhang et al., 2023b) | 0.8532 | 0.8657 | 0.0681 | 0.9211 | 0.8888 | 0.0699 |
| LIQ³A (Ours) | 0.8753 | 0.8836 | 0.0740 | 0.9322 | 0.9068 | 0.0664 |

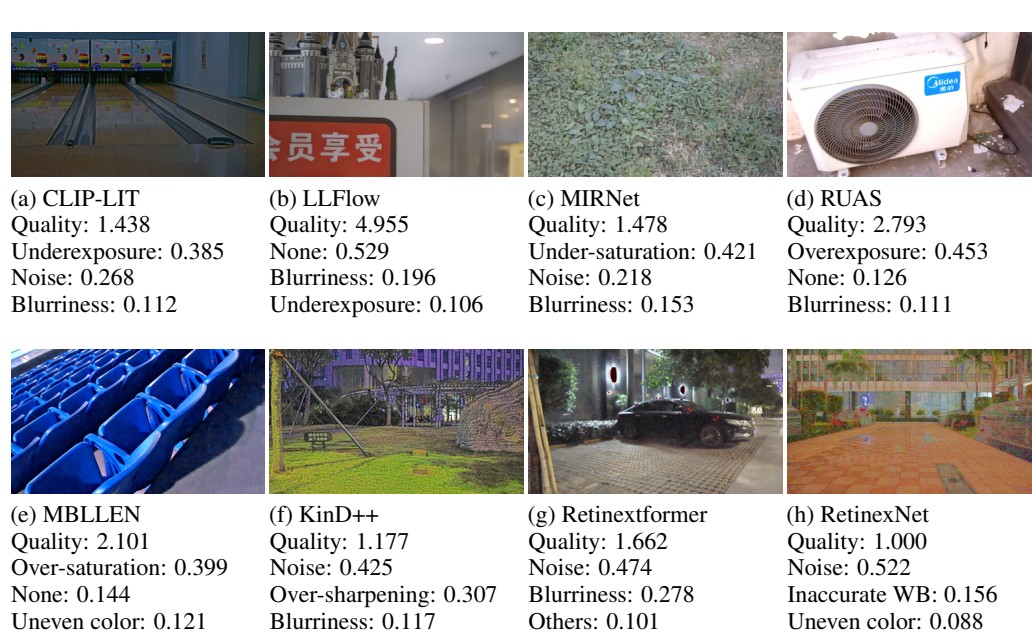

(a) CLIP-LIT
Quality: 1.438
Underexposure: 0.385
Noise: 0.268
Blurriness: 0.112

(b) LLFlow
Quality: 4.955
None: 0.529
Blurriness: 0.196
Underexposure: 0.106

(c) MIRNet
Quality: 1.478
Under-saturation: 0.421
Noise: 0.218
Blurriness: 0.153

(d) RUAS
Quality: 2.793
Overexposure: 0.453
None: 0.126
Blurriness: 0.111

(e) MBLLEN
Quality: 2.101
Over-saturation: 0.399
None: 0.144
Uneven color: 0.121

(f) KinD++
Quality: 1.177
Noise: 0.425
Over-sharpening: 0.307
Blurriness: 0.117

(g) Retinextformer
Quality: 1.662
Noise: 0.474
Blurriness: 0.278
Others: 0.101

(h) RetinexNet
Quality: 1.000
Noise: 0.522
Inaccurate WB: 0.156
Uneven color: 0.088

Figure 4: Quality scores and probabilities of primary distortions predicted by LIQ³A for enhanced images produced by different LLIE methods.

HyperIQA (Su et al., 2020), MANIQA (Yang et al., 2022), ARNIQA (Agnolucci et al., 2024b), TOPIQ (Chen et al., 2024), LIQE (Zhang et al., 2023b), and Q-Align Wu et al. (2024c). Among all competing methods, only LIQE and the proposed LIQ³A are capable of performing both quantitative quality prediction and qualitative distortion distribution estimation.

**Quantitative Results**. We list the quantitative results in Table 3, from which we have several insightful observations. First, by leveraging strong prior knowledge from an advanced multi-modal large language model (MLLM), VisualQuality-R1 outperforms other pre-trained methods across both subsets. Second, all re-trained BIQA models outperform pre-trained ones, highlighting a clear domain shift between low-light enhanced images and standard IQA datasets, and underscoring the need for task-specific fine-tuning. Third, sharing a similar design, LIQ³A outperforms LIQE, validating the superiority of the SigLIP-2 backbone over CLIP and the effectiveness of the NaFlex mechanism in enabling multi-scale representations that better capture quality-aware image features.

**Qualitative Results**. We present qualitative examples in Fig. 4 to intuitively demonstrate LIQ³A's ability to perform both quantitative and qualitative evaluations of low-light enhanced images. Thanks to our pairwise learning-to-rank training scheme across both subsets, LIQ³A effectively learns a common perceptual space in which images from Subset 1 (Fig.4(a)–(e)) and Subset 2 (Fig.4(f)–(h)) are

Table 4: SRCC and PLCC results on three datasets under the cross-dataset setup. The top two performances are highlighted in **bold**.

| Method | LIEQ | | LEISD | | Hybrid-LLIE | |
|---|---|---|---|---|---|---|
| | SRCC (↑) | PLCC (↑) | SRCC (↑) | PLCC (↑) | SRCC (↑) | PLCC (↑) |
| MUSIQ (Ke et al., 2021) | 0.7434 | 0.7411 | 0.4512 | 0.4346 | 0.4689 | 0.4566 |
| CLIPIQA (Wang et al., 2023a) | 0.4500 | 0.4664 | 0.1864 | 0.1942 | 0.4306 | 0.4486 |
| QualiCLIP+ (Agnolucci et al., 2024a) | 0.7348 | 0.7273 | 0.4849 | 0.4554 | 0.5984 | 0.5789 |
| UNIQUE (Zhang et al., 2021a) | 0.7809 | 0.7770 | 0.5346 | 0.5472 | 0.5371 | 0.5415 |
| VisualQuality-R1 (Wu et al., 2025) | **0.8487** | **0.8479** | 0.7005 | 0.7100 | **0.7120** | **0.7095** |
| DBCNN (Zhang et al., 2018) | 0.6233 | 0.6298 | 0.6136 | 0.6580 | 0.4747 | 0.4948 |
| HyperIQA (Su et al., 2020) | 0.5485 | 0.5570 | 0.5594 | 0.6247 | 0.4664 | 0.4743 |
| MANIQA (Yang et al., 2022) | 0.7247 | 0.7308 | 0.6333 | 0.6779 | 0.6400 | 0.6424 |
| ARNIQA (Agnolucci et al., 2024b) | 0.4563 | 0.4829 | 0.5569 | 0.6148 | 0.4744 | 0.4786 |
| TOPIQ (Chen et al., 2024) | 0.7130 | 0.7237 | 0.6606 | 0.6967 | 0.5052 | 0.5117 |
| Q-Align Wu et al. (2024c) | 0.8133 | 0.8007 | 0.7329 | 0.7460 | 0.6864 | 0.6898 |
| LIQE (Zhang et al., 2023b) | 0.7549 | 0.7680 | **0.7506** | **0.7919** | 0.6231 | 0.6406 |
| LIQ³A (Ours) | **0.8165** | **0.8121** | **0.7729** | **0.7979** | **0.7470** | **0.7398** |

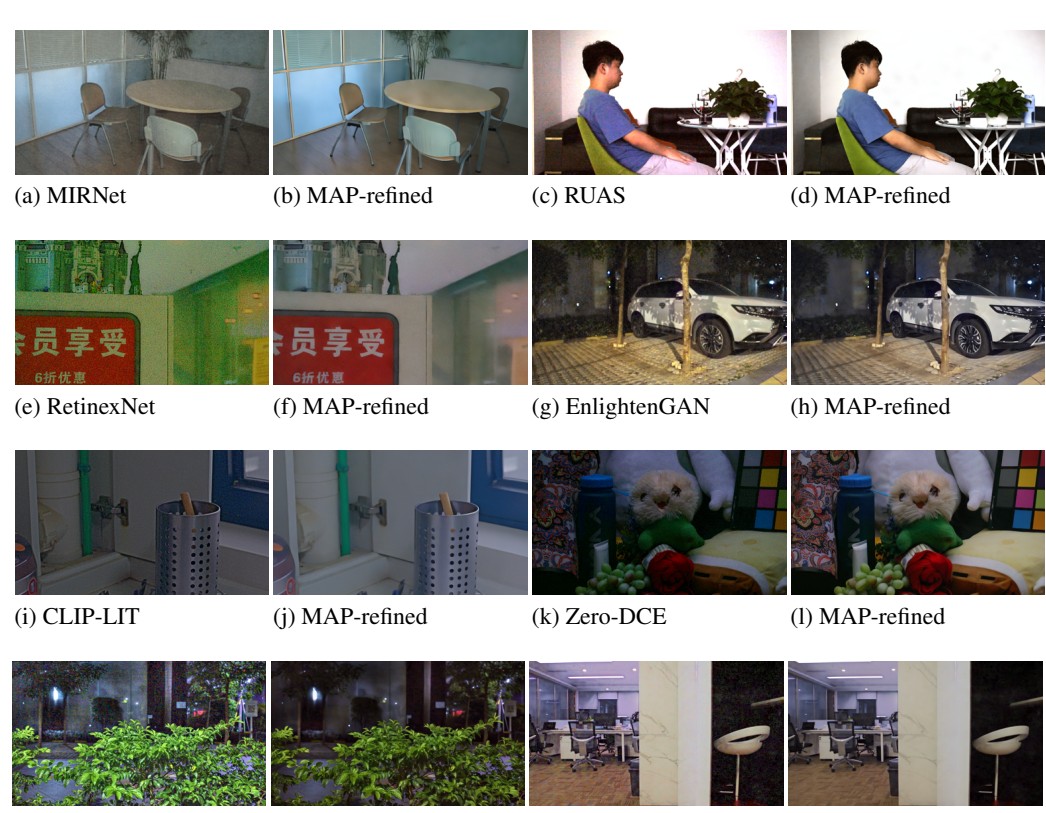

| | | | |
|---|---|---|---|
| (a) MIRNet | (b) MAP-refined | (c) RUAS | (d) MAP-refined |
| (e) RetinexNet | (f) MAP-refined | (g) EnlightenGAN | (h) MAP-refined |
| (i) CLIP-LIT | (j) MAP-refined | (k) Zero-DCE | (l) MAP-refined |
| (m) AGLLNet | (n) MAP-refined | (o) MBLLEN | (p) MAP-refined |

Figure 5: Visual examples of MAP-refined images from outputs generated by different LLIE methods, where LIQ³A is employed to guide the perceptual optimization.

well aligned, even though their MOSs are not directly comparable. In addition, LIQ³A's predictions clearly indicate that Noise and Blurriness are the most dominant distortion types, which are consistent with the overall distortion distribution illustrated in Fig. 3.

**Generalizability Testing**. To evaluate the generalizability of LIQ³A, we perform cross-dataset testing on LIEQ Zhai et al. (2021) and LEISD Lin et al. (2023), corresponding to diverse contents and LLIE algorithms. In addition, we select five low-light images from each of DICM (Lee et al., 2013), MEF (Ma et al., 2015), LIME (Guo et al., 2016), NPE (Wang et al., 2013), and VV (Vonikakis et al., 2018), and enhance them with the same 21 algorithms used in LIDQ, resulting in 550 images. We perform formal subjective testing to annotate their perceptual quality. The resulting dataset,

Table 5: Performance comparison of different loss weights on Subset 1 and Subset 2. ↑ means higher is better, ↓ means lower is better.

| Loss Weights | | | | Subset 1 (LOL-v1 & LOL-v2) | | | Subset 2 (SDSD) | | |
|---|---|---|---|---|---|---|---|---|---|
| Model | $\lambda_f$ | $\lambda_p$ | $\lambda_d$ | SRCC (↑) | PLCC (↑) | EMD (↓) | SRCC (↑) | PLCC (↑) | EMD (↓) |
| I | 1 | 1 | 0 | 0.8660 | 0.8730 | 0.1742 | 0.9304 | 0.8800 | 0.2853 |
| II | 1 | 0 | 0 | 0.8612 | 0.8729 | 0.2064 | 0.9281 | 0.8533 | 0.1814 |
| III | 0 | 1 | 0 | 0.8621 | 0.8763 | 0.1892 | 0.9292 | 0.9002 | 0.2473 |
| IV | 0 | 0 | 1 | -0.5221 | 0.5423 | 0.0753 | -0.4239 | 0.4142 | 0.0664 |
| V | 1 | 1 | 1 | **0.8753** | **0.8836** | **0.0740** | **0.9322** | **0.9068** | **0.0664** |

termed Hybrid-LLIE, is used to evaluate the cross-scene generalizability of IQA methods. The results are reported in Table 4, from which we have three primary observations. First, in contrast to other competing methods, VisualQuality-R1 and Q-Align demonstrate strong performance across all three datasets, highlighting the generalizability of MLLM-based IQA models. Second, with far fewer parameters, LIQ³A attains comparable or superior cross-dataset performance to MLLM-based methods, validating the soundness of our design. Third, adhering to a similar design philosophy, LIQ³A consistently surpasses LIQE on all three datasets, confirming that using the advanced vision–language model SigLIP-2 as the backbone, rather than designing a new model from scratch, delivers substantial improvements in generalization.

**Ablation Study**. We perform ablation studies to assess the contribution of each loss term by varying their corresponding weights. As summarized in Table 5, several insights can be drawn. First, training with only $\ell_f$ (Model II) or only $\ell_p$ (Model III) achieves comparable SRCC results on both subsets. Model III attains higher PLCC on Subset II, demonstrating its effectiveness in enhancing prediction precision under diverse luminance conditions. Second, models trained without $\ell_d$ (Models I–III) fail to generate meaningful distortion–distribution estimates, highlighting the necessity of explicit supervision for learning qualitative degradation characteristics. In contrast, relying solely on $\ell_d$ (Model IV) is insufficient for quantitative quality prediction. Finally, incorporating all three losses (Model V) results in accurate qualitative distortion–distribution estimation along with slight improvements in quantitative metrics, indicating beneficial knowledge transfer between the two tasks.

**Perceptual Optimization**. Beyond serving as a performance measure for LLIE, it is also highly beneficial to explore the use of a quality metric for perceptual optimization. We plug LIQ³A into the maximum a posteriori (MAP) estimation within the diffusion latents framework (Zhang et al., 2025) to perform post-enhancement on the outputs generated by LLIE methods. To evaluate this, we show in Fig. 5 image pairs, consisting of the outputs from different LLIE methods and our MAP-refined results. Although the refined images may not fully attain ideal perceptual quality, they nonetheless exhibit clear and consistent improvements over the original enhancements. These results indicate that LIQ³A integrates effectively within the MAP estimation framework, mitigating distortions introduced by diverse LLIE methods and demonstrating a reliable understanding of degradation factors in an analysis-by-synthesis manner (Grenander & Miller, 2007).

# 6 CONCLUSION AND LIMITATIONS

**Conclusion.** In this work, we conduct a comprehensive quality assessment study for low-light enhanced images. We first present LIDQ, a new dataset containing 5,566 enhanced images produced by 21 modern LLIE methods. Both quantitative and qualitative quality annotations, in the form of MOSs and distortion distributions, respectively, are obtained via well-controlled subjective testing. Building on LIDQ, we develop LIQ³A, a quality metric that effectively quantifies overall image quality and estimates distortion distributions in low-light enhanced images. We believe our dataset and metric will drive progress in both the development of LLIE methods and the advancement of LLIE evaluation metrics.

**Limitations.** First, despite being the largest LLIE IQA dataset in the literature, LIDQ remains limited in size, indicating the need for larger and more diverse datasets. In addition, although LIDQ involves 34 subjects, expanding the participant pool could further reduce subjective bias. Moreover, the current coverage of 21 LLIE methods is not exhaustive and may introduce distortion bias. Another limitation is LIQ³A's computational cost, which limits real-time or edge deployment. Lastly, integrating MLLMs for joint quantitative–qualitative assessment is a promising future direction.

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
