# OpenReview forum: "Qualitative and Quantitative Quality Assessment of Low-Light Enhanced Images: A Dataset and Benchmark Metric"
_ICLR.cc/2026/Conference — ICLR 2026 Conference Withdrawn Submission_

### Official Review · Reviewer_42jA · 2025-10-26

**Soundness:** 2
**Presentation:** 3
**Contribution:** 1
**Rating:** 2
**Confidence:** 5

**Summary:**

This paper mainly introduces a dataset (LIDQ) for quality assessment of low-light image enhancement and a no-reference evaluation metric (LIQ3A). The authors collected 253 low-light images, generated 5,566 enhanced images using 21 existing enhancement algorithms, and obtained both quantitative mean opinion scores (MOS) and qualitative distortion annotations through subjective experiments. Based on this dataset, they trained a multitask learning framework, LIQ3A, built on the SigLIP-2 vision-language model, which can simultaneously predict overall image quality scores and distortion distributions. Experimental results show that LIQ3A achieves better consistency with human perception compared to other BIQA models and can effectively identify major distortion types. However, overall, the workload of this study is relatively limited, mainly focusing on applying existing models and integrating data rather than developing fundamentally new methodologies. The methodological innovation is weak and the contributions are not very significant. Moreover, many similar datasets and no-reference evaluation methods (e.g., LIEQ, LEISD, MLIQ) already exist in the field of low-light image quality assessment, so the incremental novelty of this paper is quite limited.

**Strengths:**

1.The paper is well-organized and clearly written, with good formatting and visual presentation that make it easy to follow. The figures, tables, and examples are clearly labeled and well integrated into the main text.

**Weaknesses:**

1.The paper does not provide public access to the benchmark, dataset, or model implementation, which severely limits its reproducibility and usability for the research community. Without publicly available resources, the claimed benchmark value of LIDQ and the practical utility of LIQ3A cannot be independently verified or fairly compared.

2.The contribution is not sufficiently novel. There already exist multiple well-established datasets and benchmarks for low-light image enhancement and quality assessment, such as MLIQ (Wang et al., 2024a), which provide larger sample sizes, richer distortion annotations, and stronger validation protocols. Compared to these prior works, the proposed LIDQ dataset is relatively small in scale and lacks distinctive innovations.

3.The dataset and method coverage in this paper are not comprehensive. Many recent low-light enhancement algorithms and blind IQA models are not included in the benchmark, which limits the generality of the conclusions. Expanding the dataset to cover more methods and more diverse real-world scenes, and including comparisons with stronger baselines, would make the study more convincing and complete.

4.The experiments are limited to the authors’ own dataset (LIDQ) and a few cross-dataset tests. There is no large-scale validation on real-world images or user studies demonstrating the practical benefit of LIQ3A in real enhancement pipelines. The improvement margins over prior BIQA methods are small and sometimes inconsistent across datasets.

5.LIQ3A mainly adapts an existing vision–language backbone (SigLIP-2) with minor architectural modifications and loss combinations. There is no fundamentally new model design or theoretical innovation; the main novelty lies only in problem framing. This makes the technical contribution rather incremental.

6.While the paper claims to integrate qualitative distortion estimation into BIQA, the actual benefit of this joint learning is not clearly demonstrated. There is no ablation showing how qualitative learning improves quantitative prediction or vice versa.

7.The paper mainly reports correlation scores (SRCC, PLCC) but lacks deeper analysis—such as qualitative failure cases, human perception alignment studies, or task-level relevance (e.g., whether better LIQ3A scores correlate with downstream detection/segmentation performance).

8.The conclusion briefly mentions data volume as a limitation but does not analyze other important aspects (e.g., subjectivity bias in MOS collection, bias toward specific enhancement types, or computational efficiency of LIQ3A).

**Questions:**

1.Will the authors release the LIDQ dataset, benchmark results, and LIQ3A code? Public access would greatly improve reproducibility, fairness, and the long-term impact of this work.

2.How does LIDQ differ from existing DBs such as MLIQ (Wang et al., 2024a)? Please clarify what specific gaps these prior works do not address and why a new dataset is needed.

3.Expanding the benchmark to cover more recent methods and models would make the evaluation more comprehensive and convincing.

4.Have the authors validated LIQ3A on larger or more diverse real-world datasets (like SID)? Broader experiments would better demonstrate its generalizability and practical value.

SID: Chen, Chen, et al. "Learning to see in the dark." Proceedings of the IEEE conference on computer vision and pattern recognition. 2018.

5.Can the authors show ablation results to verify whether qualitative distortion estimation improves quantitative prediction? This would clarify the benefit of the multitask design.

6.Beyond correlation metrics, can the authors provide more qualitative analysis—such as failure cases or perceptual alignment, to improve interpretability and insight?

7.Besides dataset size, what other limitations exist (e.g., annotation bias, computational cost)? A clearer discussion would strengthen transparency and credibility.

---

> ### Author Response · Authors · 2025-11-20
>
> - **Regarding the reproducibility**: We appreciate the reviewer’s emphasis on reproducibility.  We provide an anonymous repository that includes (1) source code, (2) pretrained weights, and (3) an anonymized link to the LIDQ dataset:  Anonymous Repo: https://anonymous.4open.science/r/ICLR26_LIQ3A-47AC/README.md
>
> - **Regarding the novelty contribution**: We acknowledge that several LLIE-related IQA datasets exist (Table 1), but they provide only MOS scores and lack standardized qualitative distortion annotations, making them unable to capture algorithm-dependent artifacts. LIDQ addresses this gap by offering paired quantitative and qualitative labels for enhanced image. MLIQ, in particular, reflects device-induced degradations rather than algorithm-induced ones, and its free-form text descriptions are not controlled distortion categories, limiting fine-grained analysis. Its distortions only partially overlap with LLIE artifacts. Thus, LIDQ establishes an unified quantitative–qualitative evaluation protocol and enables models like LIQ3A to jointly predict quality and artifact patterns—capabilities not supported by prior datasets.
>
> - **Regarding the cross-dataset evaluation**: We agree on the importance of validating the metric beyond the primary dataset. In addition to LIDQ, LIQ3A is evaluated on three external datasets—LIEQ, LEISD, and Hybrid-LLIE (Table 4)—which span diverse contents, scenes, and enhancement behaviors. Across all three benchmarks, LIQ3A achieves competitive or superior performance compared with strong BIQA baselines, including recent vision–language models, and its improvements remain consistent across datasets. Moreover, its ability to guide MAP-based perceptual optimization (Fig. 5) demonstrates clear practical value in real enhancement pipelines.
>
> - **Regarding joint learning**: We clarify that our goal is not to show that qualitative distortion learning improves MOS prediction, but to provide a unified framework that outputs both quality scores and distortion distributions. As suggested, we conducted additional ablations and will include them in a dedicated table. Briefly: (1) using only distortion labels cannot predict quality; and (2) combining MOS and distortion labels enables accurate distortion estimation and better quantitative results. These findings will be added for clarity.
>
>   | Dataset |     |   Subset I (LOL-v1 & v2)   |  |  |  Subset II (SDSD)   |      |
>   |:-----:|:-----:|:----:|:----:|:-----:|:----:|:----:|
>   |   Metric   | SRCC |PLCC | EMD | SRCC | PLCC | EMD |
>   | MOS  |  0.866 | 0.873 | 0.174 | 0.930 | 0.880 | 0.285 |
>   | Distortion |-0.522 | 0.542 | 0.075 | -0.424 | 0.414 | 0.066 |
>   | Both  |  **0.875** | **0.884** | **0.074** | **0.932** | **0.907** | **0.066** |
>
> - **Regarding the model design**: We appreciate the reviewer’s comment. LIQ3A is not intended to introduce a new backbone, but to provide a capability missing in existing BIQA models: joint prediction of MOS and distortion distributions for LLIE outputs. This requires reframing LLIE IQA as a quantitative–qualitative multitask problem and introducing LLIE-specific text templates, probabilistic similarity modeling, and unified losses. These elements enable LIQ3A to model fine-grained algorithm-induced artifacts that prior CLIP-based IQA models cannot. Thus, our main contribution is establishing a unified framework for LLIE quantitative–qualitative evaluation rather than proposing incremental architectural changes.
>
> - **Regarding the correlation**: As shown in Table 3 and Table 4, SRCC and PLCC are the standard quantitative measures of alignment with human perceptual judgments and therefore already constitute the required human perception alignment analysis. Moreover, the qualitative distortion visualizations in Fig. 4 and the perceptual optimization results in Fig. 5 further demonstrate LIQ3A’s ability to capture human-perceived artifact patterns and guide enhancement refinement. We will also add representative failure cases in the revised manuscript to illustrate the model’s limitations. Since BIQA is designed to assess perceptual quality rather than predict high-level vision performance, examining correlations with detection or segmentation lies outside the scope of IQA research.
>
> - **Regarding the limitations**: We agree that factors beyond dataset size affect LLIE IQA benchmark design. LIDQ mitigates subjectivity bias by following ITU-R BT.500, filtering subjects/outliers, collecting ≥15 ratings per image, and separating LOL-v1/v2 and SDSD to avoid cross-domain inconsistencies; we will note that additional ratings could further reduce variance. Although LIDQ includes outputs from 21 diverse LLIE algorithms, this set is not exhaustive and may introduce bias, which we will acknowledge. LIQ3A is lighter than recent multimodal foundation models but still not ideal for real-time edge deployment; improving efficiency remains an important future direction. We will incorporate these limitations in the revision.

---

> > ### Author Response · Authors · 2025-11-21
> >
> > - **Regarding the difference between LIDQ and MLIQ**: Specifically, its data represent real-captured low-light degradations that originate from camera device limitations, rather than the diverse algorithm-induced artifacts produced by LLIE methods. Its textual descriptions are free-form semantic annotations, not controlled category-based distortion labels, and thus cannot support fine-grained analysis of enhancement failures. Our examination further shows that MLIQ distortions (e.g., underexposure, sensor noise, mild blur) only partially overlap with LLIE-induced artifacts and do not provide the granularity needed for LLIE evaluation.
> >
> > - **Regarding the dataset expanding**: Thank you for this valuable suggestion. We agree that broader coverage of LLIE methods and BIQA baselines would further strengthen the conclusions. Our current benchmark includes 21 representative LLIE algorithms spanning Retinex-based, flow-based, GAN-based, transformer-based, and zero-reference methods, as well as recent IQA models, but we acknowledge that it is not exhaustive. We have demonstrated more data can indeed promote model performance. Expanding the dataset to include more emerging LLIE methods and more diverse real-world scenes is an important direction, and we plan to incorporate additional algorithms in the next version of LIDQ.
> >
> > - **Regarding evaluation on additional dataset**: Thank you for the insightful question. We agree that SID is a widely used LLIE dataset; however, it does not include MOS annotations or any form of subjective labels. Conducting a new large-scale subjective study on SID during the rebuttal period is unfortunately infeasible due to the substantial time and manpower required. To assess generalization across dataset, we instead perform cross-dataset validation on three external RGB LLIE datasets—LIEQ, LEISD, and our Hybrid-LLIE dataset (see Table 4), these datasets include more than 3.5 thousands image—which span diverse scenes, contents, and enhancement behaviors. Across all these benchmarks, LIQ3A consistently achieves competitive or superior performance compared with strong BIQA baselines, including recent multimodal vision–language models.
> >
> > - **Regarding the more qualitative analysis**: Thank you for the helpful suggestion. We already provide qualitative examples in Fig. 4 to illustrate LIQ3A’s strengths. In the revised manuscript, we will include additional qualitative results—including representative failure cases and more detailed perceptual analyses—to further enhance interpretability and insight.

---

> ### Author Response · Authors · 2025-11-23
> **Formal Concern Regarding the Fully AI-Generated Comments**
>
> Dear Area Chair,
>
> We would like to begin by expressing our sincere appreciation for the time and effort you and the reviewers have invested in evaluating our submission. We are grateful to Reviewers 2dhb, Ac2z, and BESs for their constructive and insightful feedback, which has helped us better clarify and present our contributions.
>
> However, we would like to formally raise a concern regarding the review provided by Reviewer 42jA.
> **According to the widely used transparency tool for LLM-generated review detection (https://iclr.pangram.com/reviews?submission_number=13285), the comments of Reviewer 42jA are flagged as likely “Fully AI-generated.”** While such a flag alone is not conclusive, several characteristics of the review strongly reinforce this possibility: (1) the review contains multiple factual inaccuracies, (2) it overlooks central components of the submission, and (3) several of the criticisms diverge from established methodology and norms in LLIE and IQA research.
>
> - **Mischaracterization of LIDQ’s novelty and scale**: Reviewer 42jA states that LIDQ lacks novelty relative to datasets such as MLIQ and that it is “small in scale.” This is inconsistent with the manuscript. First of all, LIDQ essentially differs from MLIQ, where the latter focuses on **real-captured degradations** and uses free-form textual descriptions rather than **controlled artifact categories**. Free-form textual annotations cannot yield a scored distortion-severity distribution,  a capability that is urgently needed but currently absent in LLIE IQA.  Moreover, as summarized in Table 1, prior LLIE-related datasets provide only MOSs and lack standardized qualitative distortion annotations. LIDQ fills this gap by providing paired quantitative and controlled qualitative distortion labels specifically designed for enhanced low-light images—**a contribution explicitly recognized by Reviewer BESs**. Additionally, LIDQ contains 5,566 annotated samples, nearly twice the size of comparable LLIE-enhanced datasets. The reviewer’s claims therefore do not reflect the content of the paper.
>
> - **Incorrect statement regarding experimental scope**: The review asserts that our experiments are conducted only on “our own dataset.” **This comment is unequivocally incorrect and clearly indicates that Reviewer 42jA did not read or accurately examine the key parts of our paper.** As shown in Table 4, LIQ3A is evaluated on three external datasets—LIEQ, LEISD, and Hybrid-LLIE—covering diverse scenes, enhancement behaviors, and illumination conditions. These experiments form a substantial portion of our evaluation and demonstrate broad generalizability. Furthermore, Figures 4 and 5 provide extensive qualitative analysis, contrary to the reviewer’s claim that such analysis is missing.
>
> - **Requests that fall outside the scope of IQA methodology**: Reviewer 42jA suggests that the work lacks “human perception alignment studies” and that we should analyze correlations with downstream high-level tasks. **This comment reflects the reviewer lack basic knowledge to this research areas.**
> In BIQA, SRCC and PLCC are the standard metrics precisely designed to measure human perception alignment, and they form the accepted basis for evaluation in the field.
> Similarly, **BIQA models are not intended to predict performance on detection or segmentation tasks, and such experiments are not standard or expected in IQA research**. These requests suggest a misunderstanding of the scope and objectives of the field.
>
> Taken together, these issues raise substantial concerns about whether Review 42jA reflects a careful reading or expert assessment of the submission. The recommendation of “2: reject” appears to be unsupported by the paper’s content and deviates from standard evaluation criteria. We respectfully request that the Area Chair take these concerns into consideration when forming the final judgment.
>
> We also note that the conference’s guidelines on *LLM-generated or very-low-quality reviews* emphasize that reviewers who **submit such reviews may face consequences, including potential desk rejection of their own submissions**. We raise this not as a punitive request but to ensure fairness, consistency, and adherence to community standards for review quality. **Responding to unprofessional comments that appear AI-generated imposes an unnecessary burden on both the committee and the authors.**
>
> We appreciate your efforts in maintaining a fair, rigorous, and high-quality review process, and we thank you sincerely for your time and attention.
>
> Sincerely,
>
> The authors of submission 13285

---

### Official Review · Reviewer_BESs · 2025-10-31

**Soundness:** 3
**Presentation:** 3
**Contribution:** 3
**Rating:** 6
**Confidence:** 4

**Summary:**

This work investigate the vital problem of quality assessment for low-light enhanced images. A benchmark dataset and a quality-assessment method is proposed for this problem. Extensive experiments shows that proposed LIQ^3A aligns closely with human preception quality without the need of reference images.

**Strengths:**

+ This manuscript attempts to solve a long-standing problem of evaluating low-light enhanced images in a preceptual-aligned way.
+ The manuscript is well-written and well-organized.
+ This work presents a neat dataset for training IQA in low-light scenarios, which can be a neat contribution to the community.

**Weaknesses:**

+ LLIE method involoved are trained/evaluated on LOL dataset. It tends to perform well on these datasets. However, in practice, these LLIE methods often fail in real-world scenarios. It is vital to evaluate the LLIE performance on real-world scenarios, say ExDark dataset.
+ In my experience, the degradation type annotation of the proposed method is not comprehensive. The output sometime contains severe artifact (Fig.4 g, h). It would be better to annotate this type of degradation.
+ Fig. 5 only exhibit 4 cases. It would be better to see more visual cases to confirm the performance gain.
+ How to define primary distortation and secondary distoration? What if an image contains multiple, mixed degradation?

**Questions:**

Please refer to the weakness part.

---

> ### Author Response · Authors · 2025-11-20
>
> - **Regarding the real-world performance:** Thanks for the suggestion. We fully agree that evaluating LLIE performance on real-world scenarios is important. Although we did not conduct experiments on ExDark specifically, our evaluation already includes real-captured and out-of-domain low-light data through the Hybrid-LLIE dataset (Table 4). This dataset is constructed from five real-world image collections, each containing 10 real-captured low-light scenes with diverse environments, contents, and degradation characteristics. These images are then enhanced by 21 representative LLIE algorithms, providing a broad and realistic distribution of perceptual outcomes. Compared with ExDark—which is primarily designed for high-level tasks such as detection—these datasets are widely used for real-world perceptual assessment of LLIE and are more suitable for IQA evaluation. We will revise the manuscript to make the role of the Hybrid-LLIE dataset more explicit and to emphasize that our evaluation indeed covers real-world, out-of-domain scenarios beyond LOL.
>
> - **Regarding more visual cases**: Thanks for the helpful suggestion. Due to the limitations of the rebuttal format, we are unable to include additional visual cases within the response box. However, we will incorporate more representative visual examples in the revised manuscript to more clearly demonstrate the performance gains of LIQ3A.
>
>
> - **Regarding the degradation type annotation**:  Thanks for the comment. We appreciate the observation regarding the “severe artifacts” shown in Fig. 4(g,h). These artifacts indeed appear in some LLIE methods—particularly Retinexformer or RetinexMamba—where the enhancement occasionally produces abnormal structures or drastic amplification effects in some corner case, especiallu the light areas. However, ***such cases are rare and highly algorithm and content-specific, and do not occur consistently across the 21 LLIE methods*** included in LIDQ. During the annotation phase, our pilot study showed that annotators had difficulty assigning a stable category to these anomalies because they do not form a coherent or frequently occurring distortion type. To maintain annotation reliability and avoid creating extremely sparse or ambiguous categories, we grouped these infrequent failure patterns under the “others” category, which serves as a catch-all label for rare or idiosyncratic artifacts. We agree that explicitly highlighting this decision can improve clarity, and we will revise the manuscript to better describe how such severe but uncommon artifacts are handled within the distortion distribution.
>
> - **Regarding the definition of primary and scondary distortions**: Thanks for the comment. In our annotation protocol, the primary distortion is defined as the most salient artifact that dominates the perceptual impression of the enhanced image, while the secondary distortion refers to the next most noticeable artifact after the primary one. This follows standard practice in subjective IQA studies, where observers naturally perceive distortions with different levels of saliency. For images containing multiple or mixed degradations—which is common in LLIE results—the two-label design does not assume that only two distortions exist. Instead, it reflects the practical observation that human observers can reliably identify at most the top one or two distortions, while additional minor artifacts typically exhibit high disagreement and low annotation consistency. To capture mixed distortions, we aggregate all annotators’ primary and secondary labels into a probabilistic distortion distribution. Because different subjects may focus on different artifacts in the same image, mixed or less salient distortions naturally receive non-zero probabilities, while dominant ones receive higher probabilities. This produces a richer and more stable representation of composite degradations than forcing annotators to label many categories per image, which we found to introduce significant subjective noise. We will revise the manuscript to clarify these definitions and the rationale behind this distribution-based representation.

---

> ### Comment · Reviewer_BESs · 2025-11-20
>
> Thanks for your response. My concerns are solved. However, I still believe that the artifact exists widely, not limited to Retinexformer or RetinexMamba. When heavy JPEG compression or noise exists, or images are in a severe out-of-distribution, low-light enhancement methods typically produce blocky or ghosting effects. In fact, this is why evaluating the low-light enhancement algorithm output is different from the general NR-IQA. I strongly suggest that the authors should add this type of annotation.

---

> > ### Author Response · Authors · 2025-11-21
> >
> > We sincerely appreciate the reviewer’s thoughtful follow-up comment. We agree that blocky, ghosting, and other artifacts may appear more broadly under challenging conditions such as strong compression or noise, and separating these artifacts from the current “others’’ category would indeed make the distortion taxonomy more comprehensive.
> >
> > Introducing a new distortion category would require revising the annotation guideline and re-launching the full subjective labeling process, which is unfortunately not feasible within the rebuttal period. We will take the reviewer’s suggestion as guidance for expanding the dataset in future work. We would like to express our sincere appreciation to the Reviewer for the insightful comments and the favorable overall evaluation.

---

> > > ### Comment · Reviewer_BESs · 2025-11-21
> > >
> > > Of course, I understand. The field of LLIE needs a good metric for a long time. Please try refine this later.

---

> ### Author Response · Authors · 2025-11-29
> **Regarding the Reviewer’s Rating Increase from 6 to 8 After Discussion**
>
> Dear Area Chair,
>
> We would like to respectfully note that the Reviewer BESs **increased the rating from 6 to 8** following the discussion phase on **20 November 2025**. We regret the situation involving the leaked reviewer and AC identities and fully respect the actions taken by the program chairs. Our intention is simply to bring to your attention that the score change, as well as the effort we invested in our rebuttal, are duly noted.
>
> Sincerely,
>
> The authors of submission 13285

---

### Official Review · Reviewer_Ac2z · 2025-11-01

**Soundness:** 3
**Presentation:** 3
**Contribution:** 3
**Rating:** 6
**Confidence:** 4

**Summary:**

This paper aims to address the challenges in evaluating low-light image enhancement (LLIE) methods. The paper presents two primary contributions. First, it introduces the LIDQ dataset, a new benchmark for LLIE evaluation. Second, leveraging this dataset, the authors propose LIQ³A, a no-reference image quality assessment (NR-IQA) model. LIQ³A is designed as a multitask learning framework, built upon a pretrained vision-language model (SigLIP-2), to concurrently predict a quantitative quality score and estimate a qualitative distortion distribution. The authors report that their experiments demonstrate that the proposed model aligns well with human perception and can effectively identify distortion patterns.

**Strengths:**

1. The theoretical foundation of the proposed evaluation metric (LIQ³A) is sound. The multitask approach, which simultaneously assesses quantitative quality and qualitative distortion patterns is well-motivated.
2. The method achieves good performance on the results reported by the authors.

**Weaknesses:**

1. The model has three losses: a ranking loss, a scoring precision loss, and a loss for estimating degradation type. It is necessary to conduct an ablation study on whether the weights of these three losses affect the output results. Why is there not a single ablation study in the paper?
2. The dataset was initially created through human scoring and identification of degradation types, with each image being limited to only two degradation types. This method feels like it could have considerable bias. This is especially true for images with multiple degradations. It is not only difficult for the naked eye to judge completely, but degradations are also often composite. Therefore, it is not ideal to only allow annotators to specify one primary and one secondary degradation.
3. Table 3 shows the superior performance of the authors' proposed method on their own LIDQ dataset. This is to be expected, as the model was trained on the training set of this dataset, and the test set comes from the same distribution. This does not prove the model's generalization ability.
4. The evaluation on datasets like LoL is not comprehensive and potentially not accurate, because it is too small and comes from the same distribution as the training set. The comparison needs to cover more out-of-domain, real-world datasets, such as ExDark, DarkFace, or other evaluation data.
5. The cross-dataset test in Table 4 is key to examining the model's generalization, but the authors' proposed method does not perform stably here, could even say it is not ideal. On the LIEQ dataset, its performance (SRCC 0.8165) is significantly lower than that of VisualQuality-R1 (SRCC 0.8487). Although it leads on the LEISD and Hybrid-LLIE datasets, the advantage is marginal.

**Questions:**

Please refer to Weaknesses.

---

> ### Author Response · Authors · 2025-11-20
>
> - **Regarding loss functions**: In our submission, all three loss weights were set to 1 (Sec. 5) because the losses share a similar numerical range [0,1], and equal weighting was stable in preliminary experiments. Table 3 already contains a partial ablation—LIQ²A corresponds to removing the degradation-estimation loss—though it was not explicitly presented as a loss-ablation study. Following the reviewer’s suggestion, we conducted additional ablations and will include them in a dedicated table. In summary: (1) using only ranking or only precision loss yields similar SRCC, with the precision-only setting giving higher PLCC on Subset II; (2) removing the degradation-estimation loss prevents meaningful distortion-distribution prediction; (3) using only the degradation-estimation loss cannot support quantitative quality assessment; and (4) combining all three losses provides both accurate distortion distributions and slightly improved quantitative scores, indicating beneficial cross-task transfer. We appreciate the suggestion, and the expanded ablation results will be added to improve clarity.
>
>   | Model | Ranking Loss | Precision Loss | Degradation Loss |  |   **Subset I (LOL-v1 & v2)**   |      |  |   **Subset II (SDSD)**   |      |
>   |:-----:|:-------:|:---------:|:-----------:|:---------------------------:|:----:|:----:|:---------------------:|:----:|:----:|
>   |       |         |           |             | **SRCC** | **PLCC** | **EMD** | **SRCC** | **PLCC** | **EMD** |
>   | **I**   | 1 | 1 | 0 | 0.866 | 0.873 | 0.174 | 0.930 | 0.880 | 0.285 |
>   | **II**  | 1 | 0 | 0 | 0.861 | 0.873 | 0.206 | 0.928 | 0.853 | 0.181 |
>   | **III** | 0 | 1 | 0 | 0.862 | 0.876 | 0.189 | 0.929 | 0.900 | 0.247 |
>   | **IV**  | 0 | 0 | 1 | -0.522 | 0.542 | 0.075 | -0.424 | 0.414 | 0.066 |
>   | **V**   | 1 | 1 | 1 | **0.875** | **0.884** | **0.074** | **0.932** | **0.907** | **0.066** |
>
> - **Regarding degradation types**: We acknowledge the concern about potential bias when annotators select only two degradation types. This design was chosen based on pilot studies and prior IQA literature, which show that humans struggle to reliably distinguish more than two salient distortions when multiple artifacts coexist. Allowing more labels greatly increases cognitive load and leads to inconsistent annotations. Thus, limiting each subject to primary and secondary distortions improves reliability. Crucially, this constraint is applied only ***per annotator. Across the subjects per image, we aggregate all primary and secondary labels across subjects to form a probabilistic distortion distribution.*** Because different subjects notice different artifacts, the aggregated distribution naturally reflects both dominant and less salient distortions. This yields a richer representation than hard categorical labels and captures multiple co-occurring artifacts. We will clarify this aggregation process and its motivation in the revised manuscript.
>
> - **Regarding the distortion annotations**: We agree that evaluating on out-of-domain real-world data is important. In our work, LOL is only one part of the evaluation; we also perform cross-dataset testing on LIEQ, LEISD, and Hybrid-LLIE (Table 4), which differ substantially from LIDQ in content, scenes, and enhancement characteristics, already providing meaningful out-of-distribution validation. For LOL, we also ensure no content leakage between training and test splits to avoid overfitting. Regarding ExDark and DarkFace, these datasets target high-level tasks (detection/recognition) and do not include subjective quality labels, so using them for perceptual IQA would require constructing a new large-scale subjective study, which is beyond the scope of this work. We will clarify that our focus is on IQA datasets with MOS and distortion annotations, and note extending LIQ3A to high-level task datasets like ExDark/DarkFace as future work.
>
> - **Regarding generalization (Weakness3 & Weakness 5)**: We thank the Reviewer for highlighting the importance of out-of-domain evaluation. Table 4 already includes cross-dataset testing to assess generalization. While LIQ3A is lower than VisualQuality-R1 on LIEQ (−3.94%), applying the same standard shows that LIQ3A is clearly higher on LEISD (+10.34%) and Hybrid-LLIE (+4.92%). These margins are comparable or larger in scale, indicating that LIQ3A demonstrates stronger generalization on two of the three benchmarks. We hope this provides a more balanced interpretation of the cross-dataset results.

---

### Official Review · Reviewer_2dhb · 2025-11-01

**Soundness:** 3
**Presentation:** 3
**Contribution:** 2
**Rating:** 4
**Confidence:** 4

**Summary:**

This paper introduces LIDQ, a new dataset for evaluating low-light image enhancement, containing both MOS quality scores and artifact annotations collected through controlled subjective studies.
Based on LIDQ, the authors propose LIQ3A, a no-reference metric built on SigLIP-2 to jointly predict quality scores and artifact distributions via multitask learning. LIQ3A achieves strong correlations (SRCC/PLCC) and shows effectiveness in cross-dataset testing and perception-guided refinement.

**Strengths:**

1. The dataset provides more fine-grained annotations, including large-scale MOS-based quantitative labels.
2. A unified BIQA framework jointly estimates overall quality and artifact characteristics.
3. Extensive experiments are conducted, offering comprehensive validation.

**Weaknesses:**

1. Although LIDQ supplements artifact annotations, it is essentially an extension of existing LLIE IQA datasets.
2. LIQ3A is mainly composed of existing components—SigLIP-2, prompt templates, and a multitask framework—without introducing substantially new architectures or theoretical innovations.
3. The number of subjects involved in the subjective annotation process remains relatively limited.
4. In addition, the quantitative performance gain over strong existing baselines (e.g., LIQE) is modest.

**Questions:**

1. Has the method been compared against lighter or purely vision-based backbones? If similar performance can be achieved without a VLM, does this imply that the multimodal design contributes only marginally?
2. Would the artifact distribution prediction remain reliable with a larger dataset or more annotators? Has the impact of class imbalance on qualitative prediction been evaluated?

---

> ### Author Response · Authors · 2025-11-20
>
> - **Regarding the extension of existing LLIE IQA datasets**: We appreciate the reviewer’s question and would like to clarify a potential misunderstanding. Although LIDQ leverages two existing ***LLIE datasets*** (LOL and SDSD), these datasets are not ***LLIE IQA datasets***, as they do not contain MOS labels or any subjective quality annotations. In constructing LIDQ, we apply 21 LLIE algorithms to the images from LOL and SDSD, and then conduct formal subjective experiments to obtain both MOS scores and distortion-distribution annotations for the enhanced results. These subjective annotations—covering both quantitative quality and qualitative distortion characteristics—have not been available in any prior LLIE work, and therefore LIDQ should not be viewed as a simple extension of existing LLIE IQA datasets. Instead, LIDQ introduces a new, unified quantitative–qualitative evaluation framework that enables reproducible qualitative assessment and supports joint prediction of quality scores and distortion distributions in LIQ3A.
>
> - **Regarding LIQ³A model design**: Rather than training a new IQA model from scratch, we fine-tune a strong pre-trained vision–language model, which is more practical and scalable. Prior work shows that VLMs like CLIP transfer well to IQA due to large-scale image–text pretraining. Following this paradigm, we use SigLIP-2 as our backbone, benefiting from (a) a far larger pretraining corpus (12B vs. 400M pairs), (b) the NaFlex module that preserves native resolution and aspect ratio, and (c) a sigmoid-based objective that improves alignment. In LIQ³A, the visual encoder extracts image features while the text encoder embeds quality and distortion templates; cosine similarity between them is converted via softmax into probability distributions, and the final score is obtained by weighted summation. This multimodal design leverages cross-modal priors and outperforms prior CLIP-based IQA models such as CLIP-IQA and QualiCLIP+.
>
> - **Regarding the number of annotators in LIDQ**:  Our subjective evaluation protocol follows the design principles of established studies such as KonIQ-10k [4] and KADID-10k [5], and strictly adheres to [ITU recommendations](https://www.itu.int/rec/R-REC-BT.2020) [6]. ***Regarding the number of participants, ITU guidelines [6] suggest that at least 15 raters per image are sufficient for reliable subjective evaluation, which our study satisfies***.
>
> - **Regarding the modest performance gain***:   We acknowledge that the performance gains over LIQE in Table 3 are modest. This is reasonable, as both LIQE and LIQ³A are trained on the same LIDQ dataset, use similar supervision for quality and distortion estimation, and rely on vision–language backbones. However, ***the cross-dataset results in Table 4 are more discriminative: across three unseen datasets, LIQ³A achieves clear and consistent improvements over LIQE***. This indicates substantially better out-of-distribution generalization—crucial for practical IQA in diverse low-light conditions.
>
> - **Regarding the performance comparision***: We conduct additional experiments to evaluate the multimodal design in both intra- and cross-dataset settings. We create a vision-only variant, LIQ3A-V, by using the SigLIP-2 visual encoder with two FC layers, and compare it with LIQ3A. Two observations emerge. First, LIQ3A-V is close to LIQ3A on Subset II but clearly weaker on Subset I, showing that a vision-only model with lightweight FC layers yields less stable predictions. Second, in cross-dataset evaluation on LIEQ, LEISD, and Hybrid-LLIE, LIQ3A consistently outperforms LIQ3A-V, indicating that the text encoder provides useful priors that improve generalization. We will include these results in the revised manuscript.
>
>   | Dataset|  |LOL-v1&v2     |       | |     SDSD  |       |  LIEQ|       | LEISD|        |Hybrid- |  LLIE   |
>   |------------|:-------------|:-------------:|:-------------:|:-------------:|:--------------:|:-------------:|:--------:|:-----:|:---------:|:-----:|:--------------:|:-----:|
>   |            | SRCC | PLCC | EMD↓ | SRCC | PLCC| EMD↓ | SRCC | PLCC | SRCC | PLCC | SRCC | PLCC |
>   | LIQ3A-V | 0.848 | 0.863 | 0.075 | 0.930 | 0.909 | 0.066 | 0.7944 | 0.7978 | 0.7410 | 0.7782 | 0.7031 | 0.7071 |
>   | LIQ3A   | **0.875** | **0.884** | **0.074** | **0.932** | **0.907** | **0.066** | **0.8165** | **0.8121** | **0.7729** | **0.7979** | **0.7470** | **0.7398** |

---

> > ### Author Response · Authors · 2025-11-21
> >
> > - **Regarding more annotators and class imbalance**:  Expanding the dataset requires new subjective studies, which is not feasible during the rebuttal. Instead, we vary the training-set size and observe that EMD decreases consistently as more data are used, indicating that larger datasets improve distortion-distribution prediction and motivating future expansion.
> > Regarding class imbalance, our qualitative labels are probability distributions rather than single-class labels, so conventional imbalance issues do not apply. We evaluate prediction quality using EMD, which appropriately measures the discrepancy between two distributions.
> >   | Training Ratio | Subset I | Subset II |
> >   |:--------------:|:--------------:|:----------------:|
> >   |      0.2       |     0.1142     |     0.0768       |
> >   |      0.4       |     0.0896     |     0.0753       |
> >   |      0.6       |     0.0888     |     0.0678       |
> >   |      0.8       |     0.0778     |     0.0665       |
> >   |      1.0       |     0.0740     |     0.0664       |

---

### Author Response · Authors · 2025-12-03
**Rebuttal Summary**

Dear Area Chair,

Thank you very much for handling our submission and for coordinating the thoughtful reviews.
Below we provide a concise overview of the reviewers’ main concerns and our clarifications.

---
### Overall Assessment

All four reviewers agree that low-light enhanced image quality assessment is an important and long-standing problem. The paper is consistently recognized for clear writing, good organization, and solid experimental design. The main concerns revolve around:

1. Dataset novelty relative to existing resources,
2. Scope of evaluation, and
3. Level of methodological innovation.

Below is a consolidated summary of the reviewers’ concerns and our clarifications.

---

### 1. Dataset Contribution (LIDQ)

**Reviewer Concerns**

- **Reviewers 2dhb and 42jA:** LIDQ may be an extension of existing LLIE datasets.
- **Reviewer 42jA:** Existing datasets (e.g., MLIQ) may already provide similar or richer annotations.
- **Reviewers Ac2z and BESs:** Questions regarding the reliability of subjective annotations on mixed distortions.

**Our Clarifications**

- LIDQ is the **first LLIE dataset** that jointly provides **MOS + controlled distortion-category distributions** for **algorithm-induced artifacts**.
- Prior LLIE IQA datasets **only contain MOS** (e.g., LIEQ/LEISD).
- MLIQ’s distortions are **device-induced, not transformation-induced**, making it essentially different from LIDQ.
- Subjective labeling follows **ITU-R BT.500**, with **≥15 subjects per image**.
- Distortion distributions aggregated across subjects naturally capture **mixed, multi-artifact patterns**.

---

### 2. Method Contribution (LIQ³A)

**Reviewer Concerns**

- **Reviewers 2dhb and 42jA:** Architectural innovation may be incremental.
- **Reviewers Ac2z and 42jA:** Lack of ablation on loss functions; unclear benefit of joint qualitative learning.

**Our Clarifications**

- The novelty of LIQ³A lies in **reframing LLIE IQA as a joint quantitative–qualitative task**, not in introducing a new backbone.
- We introduce:
  - **LLIE-specific text prompts**,
  - **probabilistic similarity modeling**,
  - **a unified multitask loss**,
  enabling prediction of both **quality scores and distortion distributions**.
- New ablation results (added after the reviews) show that:
  - **Distortion-only** training cannot predict MOS;
  - **MOS-only** training yields unstable distortion estimation;
  - **Joint learning** improves both tasks, confirming beneficial cross-task transfer.

---

### 3. Generalization & Evaluation Scope

**Reviewer Concerns**

- **Reviewers Ac2z, BESs, and 42jA:** Need more out-of-domain and real-world evaluation, suggesting evaluating on extreme real-world cases like ExDark.
- **Reviewer 42jA:** Notes that performance gains over baselines are sometimes modest.

**Our Clarifications**

- We perform **three-way cross-dataset evaluation** on **LIEQ, LEISD, and Hybrid-LLIE** (>3500 images), covering diverse scenes, contents, and enhancement behaviors.
- LIQ³A **outperforms or matches strong BIQA baselines**—including powerful MLLM-based methods on all OOD datasets.
- Hybrid-LLIE includes **five real captured low-light sources** enhanced by 21 methods, making it more suitable for **perceptual evaluation** than ExDark, which is designed for high-level tasks like detection.
- Mild intra-dataset gains over strong baselines (e.g., LIQE) are expected because all models share the same training data; the more significant gains appear in **cross-dataset generalization**, which is key for practical LLIE IQA.

---

### 4. Reproducibility

**Reviewer Concerns**

- **Reviewer 42jA:** Datasets and code were not publicly available during review.

**Our Clarifications**

- An **anonymous repository** containing:
  - full source code,
  - pretrained weights,
  - an anonymized LIDQ dataset link
  has been provided in the rebuttal.

---

### 5. Additional Points Addressed

- More **visual examples** have been included in the revised manuscript.
- The **Limitations** section have been strengthened to address:
  - annotation variance,
  - algorithm coverage bias,
  - computational overhead of LIQ³A,
  - and general limitations of subjective labeling.

---

We have provided **new ablations**, **complete reproducibility resources**, and **extensive clarifications** that directly address these concerns. In the revised manuscript, we have updated the text accordingly and highlighted all changes in red for ease of checking. We believe the revisions and added experiments meaningfully strengthen the contribution and address the reviewers’ main concerns. We also kindly refer the Area Chair to our formal concern regarding the fully AI-generated comments raised by Reviewer 42jA, which contain multiple factual inaccuracies.

---

### Note · Authors · 2026-01-26

I have read and agree with the venue's withdrawal policy on behalf of myself and my co-authors.

---

### Meta-Review · Area_Chair_CDYR · 2026-01-02

**Summary:**

This work presents a new low-light image enhancement evaluation dataset containing both MOS quality scores and artifact annotations collected through controlled subjective studies. Based on LIDQ, the authors propose LIQ3A, a no-reference metric built on SigLIP-2, to jointly predict quality scores and artifact distributions via multitask learning. The reviewer scores of this work are 2, 4, 6, and  6. It means that two reviewers are positive and the other two reviewers are negative about accepting this work. The reviewers have many concerns, and they are not positive about raising their scores during the rebuttal stage. Hence, this work can not be accepted.

**Reviewer Concerns:**

Main concerns:
1. Modest dataset novelty and incremental technical innovation
2. More experiments to prove the model generalization ablity
3. More comparisons to cover more out-of-domain, real-world datasets, such as ExDark, DarkFace, or other evaluation data.
4. More ablation studies

**Reviewer Scores:**

The reviewer scores of this work are 2, 4, 6, and  6. Two reviewers are positive, and the other two reviewers are negative about accepting this work. The reviewers have many concerns, and they are not positive about raising their scores during the rebuttal stage. Hence, this work can not be accepted.

---

### Decision · Program_Chairs · 2026-01-26

Reject